# Recovering High-Quality Host Genomes from Gut Metagenomic Data through Genotype Imputation

*Sofia Marcos,\* Melanie Parejo, Andone Estonba, and Antton Alberdi\**

Metagenomic datasets of host-associated microbial communities often contain host DNA that is usually discarded because the amount of data is too low for accurate host genetic analyses. However, genotype imputation can be employed to reconstruct host genotypes if a reference panel is available. Here, the performance of a two-step strategy is tested to impute genotypes from four types of reference panels built using different strategies to low-depth host genome data ($\approx 2 \times$ coverage) recovered from intestinal samples of two chicken genetic lines. First, imputation accuracy is evaluated in 12 samples for which both low- and high-depth sequencing data are available, obtaining high imputation accuracies for all tested panels (>0.90). Second, the impact of reference panel choice in population genetics statistics on 100 chickens is assessed, all four panels yielding comparable results. In light of the observations, the feasibility and application of the applied imputation strategy are discussed for different species with regard to the host DNA proportion, genomic diversity, and availability of a reference panel. This method enables leveraging insofar discarded host DNA to get insights into the genetic structure of host populations, and in doing so, facilitates the implementation of hologenomic approaches that jointly analyze host and microbial genomic data.

## 1. Introduction

The large molecular datasets generated through shotgun DNA sequencing regularly contain useful information to characterize taxa, functions, and structures beyond the primary aim of the study. This is especially true in metagenomic datasets that often present mixtures of DNA from eukaryotic, prokaryotic, and viral origin.[1] While primarily used for characterizing the genomic architecture of microbial communities, metagenomic data generated from intestinal contents or feces can also be used for extracting useful genomic information of the animal host.[2] In fact, hologenomic approaches, that entail joint analysis of animal genomes along with metagenomes of host-associated microorganisms to study animal-microbiota interactions, can benefit from such optimization strategies.[3,4]

However, mining host genomic data from metagenomic datasets presents a number of challenges. The fraction of host sequences in the metagenomic mixture is often unpredictable, and can range from a negligible proportion (<5%) to an almost complete representation (>95%) of the sample,[5] even within a single taxon and sample type.[6] Hence, a given amount of metagenomic sequencing effort does not ensure that the desired depth of host DNA sequencing will be reached. When the host DNA fraction in the metagenomic mixture is low, achieving the desired sequencing depth requires increasing sequencing effort, with its respective economic burden. In consequence, the amount of host DNA sequences generated is often insufficient for accurate variant calling.

One useful strategy for efficient data mining of host genomic information is genotype imputation, which consists in estimating missing haplotypes of poorly characterized genomes using a reference panel of high-quality genotypes.[7] Thus, information gaps of genomes with very low sequencing depth can be reconstructed based on the haplotype information of a properly characterized representative panel. Genotype imputation of single nucleotide polymorphisms (SNPs) is a widely employed approach in association studies to increase the density of variants of genomic datasets.[8,9] The recent generation of large high-quality genomic databases, such as the human 1000 Genomes Project (1000G)[10] and the 1000 Bull Genomes Project,[11] has improved the accuracy of imputation and increased the statistical power of association analyses, especially for rare variants.[12,13] However, ideal reference panels are only available for a limited number of model and farm species, and they require high computational capacity.

When large reference panels are not available, an alternative strategy is to create a custom panel using a representative subset

S. Marcos, M. Parejo, A. Estonba
Applied Genomics and Bioinformatics
University of the Basque Country (UPV/EHU)
Leioa, Bilbao 48940, Spain
E-mail: sofia.marcos@ehu.eus

A. Alberdi
Center for Evolutionary Hologenomics
GLOBE Institute
University of Copenhagen
Copenhagen 1353, Denmark
E-mail: antton.alberdi@sund.ku.dk

of genomes of the studied population.[14,15] This approach can be more cost-efficient because when haplotype diversity is limited, genomic information of a subset of the population can successfully impute haplotype information to the rest of the population. Moreover, the study-specific panel can be combined with individuals from public databases,[14,15] which has been previously employed in sheep,[16] pig,[17] and chicken studies.[18]

Nevertheless, in addition to the size and diversity of the panel,[19] imputation strategy may also affect the accuracy of recovered genotypes.[20] In contrast to the standard imputation method, in which low density SNP arrays are imputed to high density based on a reference panel, shallow shotgun sequenced data display particular challenges, as no genotype is known with certainty. Recently, a two-step imputation strategy for ultra low-depth coverage samples (<1×) was introduced.[21] This approach relies on updating genotype likelihoods using a reference panel before imputing the missing genotypes in order to recover a higher density of SNPs with greater confidence. It was first proposed in human population genetics as an alternative to genotyping arrays,[21] and later applied to recover ancient human genomes.[22] To the best of our knowledge, such an imputation strategy has not been implemented yet in non-model animal populations with a limited number of available samples as a reference panel.

Here, we present a straightforward approach to recover high-quality host genomes from gut metagenomic data, showcased in farm chickens. We evaluate how the reference panel composition and sample depth of coverage affects imputation performance using four panels designed according to the resources scientists studying microbial metagenomics may have access to. We first calculate imputation accuracy between imputed and true genotypes in three chromosomes using 12 validation samples for which high-depth sequencing data are also available. Then, we employ a bigger dataset of 100 individuals to impute all autosomal chromosomes and explore how the choice of the reference panel affects commonly used population genetics statistics. Aiming at facilitating its implementation in other study systems, we provide the bioinformatic pipeline and discuss suitable panels and minimal depth thresholds required for successful imputation in light of the characteristics of the study system.

## 2. Experimental Section

### 2.1. Ethical Statement

Animal experiments were performed at IRTA's experimentation facilities in Tarragona under the permit FUE-2018-00813123 issued by the Government of Catalonia, in compliance with the Spanish Royal Decree on Animal Experimentation RD53/2013 and the European Union Directive 2010/63/EU about the protection of animals used in the experimentation.

### 2.2. Target Population and Reference Panels

#### 2.2.1. Target Population

Genomic information of the target population of 100 chickens belonging to two broiler lines (Cobb500 and Ross308, hereafter simply Cobb and Ross) was generated from metagenomic DNA extracted from the cecum contents of the birds. In short, cecum content (≈100 mg) was collected right after euthanizing the animal, and preserved in E-matrix tubes with DNA/RNA Shield buffer (Zymo Research, Cat. No. BioSite-R1200-125) at −20 °C until extraction. After physical cell disruption through bead-beating using a Tissuelyser II machine (Qiagen, Cat. No. 85300), DNA extraction was performed using a custom nucleic acid extraction protocol (details explained in Bozzi et al.),[23] and sequencing libraries were prepared using the adapter ligation-based BEST protocol.[24] Paired-end 150 bp-long reads were generated on a MGISEQ-2000 sequencing platform over multiple sequencing lanes. Sequencing effort was decided based on the desired depth of the metagenomic fraction of the samples, which was the primary objective of the data generation. A preliminary screening revealed that cecum contents contained a large fraction of microbial DNA (>80–95%), and a limited relative amount of host DNA (<5–15%) (**Figure 1**A). Aiming at ≈15 GB (gigabases, ≈50 million reads) of bacterial DNA per sample, cecum samples yielded between 0.5 and 4 GB of host DNA, which was equivalent to 0.5–4× depth of coverage of the chicken genome (≈1.05 GB). Raw data will be available from European Nucleotide Archive (ENA), with BioProject accession no. PRJEB43192 (https://www.ebi.ac.uk/ena/browser/view/PRJEB43192?show = component-projects). Until the release date, data will be made available upon request.

#### 2.2.2. Reference Samples

Internal and external high-quality genome sequence data were used to create the four reference panels tested in the study. The internal reference data were generated from ileum content samples of 12 randomly selected individuals included in the target population (5 Cobb and 7 Ross), following the same procedures as explained above. In contrast to cecum samples, ileum contents contain a very large fraction (>90–95%) of host DNA, and a small representation of microbial DNA. Hence, in order to generate a comparable amount of microbial data to that of the cecum, ileum samples were sequenced aiming 100 GB/sample. This sequencing effort yielded ≈90 GB of host DNA (≈80–90× depth of chicken genome), which enabled generating a high-quality internal panel from a subset of the studied population. In addition, chicken DNA sequence data of 40 broilers (meat producers), 20 layers (egg producers), and 20 red junglefowls (RJF, wild chickens) generated by Qanbari et al. from blood samples were used as external reference data (Figure 1A).[25]

#### 2.2.3. Composition of Reference Panels

Different combinations of the internal and external reference samples were used to create the four reference panels: i) the internal panel comprised 12 animals from the target population, ii) the external panel comprised 40 animals from two broiler lines (different from the target population), iii) the combined panel combined the previous two panels, and iv) the diverse panel also contained more distant populations (Figure 1B). The

**2100065 (2 of 14)**

(a) **Populations**

| Populations | Target population | Internal reference | External reference |
|---|---|---|---|
| Tissue type | Caecum gut content | Ileum gut content | Blood |
| Host DNA (%) | ~ 5 | ~ 95 | - |
| Breeds | Cobb (N=47) Ross (N=53) | Cobb (N=5) Ross (N=7) | Br1 (N=20) Br2 (N=20) L1 (N=10) L2 (N=10) RJF (N=20) |
| Total N | 100 | 12 | 80 |

(b) **Reference panels**

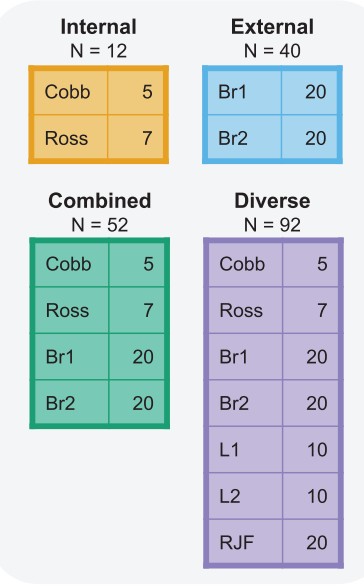

(c) **Pipeline**

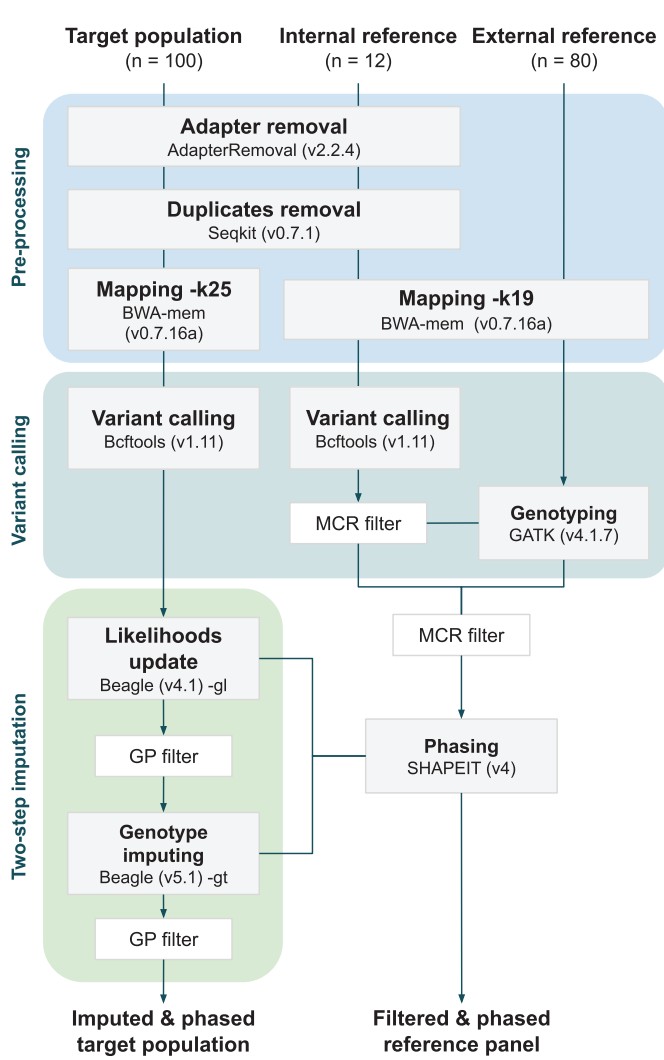

**Figure 1.** Study design and imputation pipeline for recovering host DNA. A) The characteristics of the three datasets. B) Composition and number of samples of the four reference panels used for imputation. Breeds are coded as Br1 = broiler line A, Br2 = broiler line B, L1 = white layer, L2 = brown layer, RJF = red junglefowl. C) The study design has three datasets: the target population, and internal and external reference samples. The bioinformatic procedure is divided into three steps: pre-processing, variant calling, and imputation. The input format of the starting step is a FASTQ file. After mapping we obtain a BAM file and from variant calling to the final step, procedures are performed using VCF file. The green box represents the steps proposed by Hui et al. (2020). Genotype probability (GP) filters are used during imputation and missing call rate (MCR) filters during panel design.

four panels varied in size and genomic diversity in order to see whether the composition of the reference panels affected imputation accuracy. With the internal panel, it was tested if a small subset of the target population was enough for a proper imputation in low-quality host sequence data derived from metagenomic samples. The use of just an external panel was considered to test whether it was a viable option for studies with a shortage of samples or a limited budget for high-depth host sequencing. The combined panel on the other hand, permits combining both resources. Last, the diverse panel enabled to test whether including distantly related individuals would be more effective than the three previously mentioned strategies.

### 2.3. Pipeline for Recovering Host Genotypes from Metagenomic Data

#### 2.3.1. Data Pre-Processing

All the metagenomic sequence data that contained both host and microbial DNA, were pre-processed using identical bioinformatic procedures. In short, sequencing adapters were removed using AdapterRemoval (v2.2.4)[26] and exact duplicates using seqkit "rmdup" (v0.7.1)[27] prior to the read-mapping. Read-alignment to the chicken reference genome (galGal6; NCBI Assembly accession GCF_000002315.6) was

conducted with BWA-MEM (v0.7.16a).[28] Default parameters, except for the minimum seed length (-k) were employed, which was increased to 25 in order to reduce the number of incorrectly aligned read pairs as recommended by Robinson et al.[29] This strategy was employed because the standard alignment (-k 19) presented an unconventional distribution of reads across the genome, that is, unspecified read mapping leading to regions being stacked with 80+ reads (Table S1, Supporting information). The "-M" flag, which was used to mark shorter split hits as secondary mappings were added. Aligned reads were sorted and converted into sample-specific BAM files before filtering out the metagenomic fraction (unmapped) using SAMtools view (v1.11)[30] with "-b" and "-F12" flags. Mapping statistics including depth and breadth of coverage as well as percentage of mapped reads were calculated using SAMtools' depth and flagstat functions. DNA damage of 10 cecum samples was assessed by DamageProfiler (v1.1).[31]

Pure genomic data (with no microbial fraction) generated by others[25] were downloaded from the EMBL-EBI ENA database, and mapped to the same chicken reference genome using BWA-MEM with "-k" default value and "-M" flag.

### 2.3.2. Variant Calling and Genotyping

Variants in the target population were called by chromosome with the mpileup utility of SAMtools using standard parameters (-C 50 -q 30 -Q 20). Variant calling was performed with "-m" and "-v" flags to allow variants to be called on all samples simultaneously. Raw variants were filtered using BCFtools (v1.11)[32] commands "-m2," "-M2," and "-v snps" to keep only bi-allelic SNPs.

Variants of the internal reference samples were called the same way, but additionally, low quality variants with a lower base quality than 30 (QUAL<30) and variants with a base depth higher than three times the average (DP<(AVG(DP)*3)) were removed to ensure only highly reliable variants were retained.

Since the interest was solely in imputing variants present in the target population, the external reference samples were genotyped by defining variant sites detected in the internal reference samples. Genotyping was performed for all autosomal chromosomes with GATK (v4.1.7.0)[33] HaplotypeCaller using the "–min-base-quality-score 20," "–standard-min-confidence-threshold-for-calling 30," "–alleles," and "-L" parameters to obtain calls at all given positions, followed by GATK SelectVariants "–select-type-to-include SNP" to only include SNPs.

In preliminary analyses, variants in the external panel were also called to examine the overlap with the variants present in the internal reference samples. The same procedures explained above were used for chromosome one (GGA1). Genotyping based on the positions of the internal panel and variant calling from scratch were compared by using the 40 broilers from the external panel (Figure 1B). A similar number of variants were obtained for the genotyped (2.5 M) and the variant called VCF files (2.7 M). Moreover, 28% of the variants from the 40 broilers were not present in the internal reference samples (Figure S2, Supporting information). Thus, it was decided to genotype the rest of the samples to reduce possible bias through the high number of variants specific to the external reference for the imputation of the target population.

### 2.3.3. Two-Step Imputation via Genotype Likelihood Updates

Genotypes were imputed from the four aforementioned reference panels to the target population using a two-step strategy. Prior to imputation, the reference panels were filtered by excluding variants with missing genotypes to remove any potential noise caused by inference errors, and subsequently phased using SHAPEIT (v4).[34]

Imputation was performed in two steps following Homberg et al. (2019) and Hui et al. (2020). First, genotype likelihoods were updated based on one of the reference panels using Beagle 4.1.[35] Beagle 4.1 accepted a probabilistic genotype input with "-gl" mode, and it only updated sites that were present in the input file. Second, missing genotypes in the input file were imputed using Beagle 5.1 with "-gt" mode using the same reference panel. Beagle 5.1 only accepts files with a genotype format field, like later versions than Beagle 4.1. Therefore, the latest version cannot be used for both steps. Format field genotype probabilities (GP) were generated in both steps in order to enrich confident genotypes. The highest GP was required to exceed a threshold of 0.99 after both steps using BCFtools "+setGT" plugin. The rest of the parameters were set to default. Both steps' input and output files were in VCF format. The schematic steps detailed in methods can be found in Figure 1C. Bioinformatic resources, including scripts, sample ENA accession codes and data files have been archived in the following link (10.5281/zenodo.6473506).

## 2.4. Imputation Accuracy Using 12 Validation Samples

The accuracy of the imputation using the four reference panels was tested using the 12 individuals for which both low-depth (target population) and high-depth (internal reference samples) sequence data were generated from cecum and ileum contents, respectively, hereafter referred to as validation samples. The low-depth samples of the 12 individuals had a depth of coverage spanning 0.05× to 3.73×, and breadth of coverage from 10% to 80%. For an unbiased evaluation, a leave-one-out cross-validation (LOOCV) approach was employed by excluding each of the 12 validation samples once from the reference panel in each of the different imputation scenarios. Considering the large size-variation of avian chromosomes, a macrochromosome (GGA1, 197.6 MB), a mid-size chromosome (GGA7, 36.7 MB), and a microchromosome (GGA20, 13.9 MB) were selected for the test to optimize runtime and computational resources. Concordance between the internal reference samples and imputed genotypes was calculated for each individual chicken using VCFtools, with the "–diff-discordance-matrix" option. Precision of heterozygous (het.) sites was also calculated, since these alleles were the most difficult to impute correctly. Last, imputation accuracy was estimated for variants in different minor allele frequency (MAF) bins to evaluate whether rare variants were correctly imputed. The variant frequencies were thus extracted from the internal panel by analyzing the precision of het. sites for the GGA1 in bins of 0–0.05, 0.05–0.1, 0.1–0.3, and >0.3.

## 2.5. Impact of Reference Panel on Population Genetics Inference

The implications of using different reference panels in downstream population genetics analyses were explored, including

inferences of population structure, estimates of genetic diversity, and genome scans for signatures of selection.

These analyses were run in all but two outlier samples with depths of coverage of 0.07 and 0.05×, which were below the threshold of 0.28× corresponding to the lowest successfully imputed sample in the validation set (genotype concordance of >0.90 and het. sites precision of >0.75, see results below). 100 samples were thus used (47 Cobb and 53 Ross) for which the host DNA recovery pipeline for all the autosomal chromosomes was run, and the commonly used population genetics statistics were analyzed, including observed heterozygosity (Ho), nucleotide diversity ($\pi$), pairwise distance as estimated through identity-by-state (1-IBS), and kinship estimates. The same analyses were also conducted for 10 validation samples (for the low-depth and high-depth samples) after excluding two of them, whose respective counterparts in the target populations (with 0.05 and 0.07× depth) were filtered out as mentioned above. The imputed datasets with each of the panels were filtered for missingness 0 with PLINK (v1.9).[36]

For measuring population genetics parameters, the VCF files were filtered for MAF >0.05. Ho, the percentage of het. sites over the number of variant sites, was calculated for each individual using "–het" in PLINK. $\pi$, the average pairwise sequence difference per nucleotide site, was calculated in 40 kb windows with 20 kb step size across autosomal chromosomes using VCFtools.[37] For the validation samples whole-genome windowed values were averaged to generate a genome-wide $\pi$ for each individual. For the target population, $\pi$ was calculated for each broiler line. Pairwise distance was calculated using "–distance square 1-ibs" in PLINK. Kinship was calculated with the command "–make-king square" using Manichaikul et al.'s estimator in PLINK (v2).[38]

It was further tested whether genome scans for selection between the Cobb and Ross population with each of the imputed datasets yielded consistent results. To this end, population differentiation along the genome was calculated using Weir and Cockerham's fixation index ($F_{ST}$) estimate for each panel.[39] $F_{ST}$ was calculated in sliding windows of 40 kb with 20 kb overlap across autosomal chromosomes. Window-based $F_{ST}$ values were then normalized, and regions with values above the 99th and 99.9th empirical percentile were considered as candidates for selective sweep regions.[40] The overlap of these regions across the datasets using the different reference panels was used as an estimate of consistency.

### 2.6. Statistical Analysis

Kruskal-Wallis test was performed to test for differences in average concordance across chromosomes in the 12 validation samples.[41] A paired sample $T$-test and $F$-test were performed for concordance and precision of het. sites to verify if the difference in means and variances were significant between reference panels.[42] $T$-test $p$-values were adjusted using Bonferroni's correction method.[43] Paired sample $T$-tests were performed for Ho and $\pi$ estimates in the 100 chicken population. Mantel test was performed with the R package ade4 to test the correlation between the resulting matrices from the pairwise distance and kinship analyses.[44]

## 3. Results

### 3.1. Alignment and Coverage

The mapping statistics of the 100 samples used to characterize the target population (cecum content), and the 12 internal reference samples (ileum content) were drastically different. Cecum samples showed an average of $1.84 \pm 2.35×$ (mean ± SD) depth of coverage and $52.41 \pm 24.20\%$ breadth of coverage. Ileum samples had $92.70 \pm 7.64\%$ of host DNA and an average depth of $93.16 \pm 9.07×$, practically covering the entire reference genome ($98.89 \pm 0.01\%$).

### 3.2. Imputation Accuracy of 12 Validation Samples

The internal (I, n = 12), external (E, n = 40), combined (C, n = 52), and diverse (D, n = 92) reference panels were used to study (i) the effect of panel size and diversity, and (ii) sample depth of coverage threshold on imputation accuracy in three chromosomes with contrasting dimensions. Variant calling in the internal reference samples detected 2.4 м, 470 K, and 182 K putative SNPs in chromosomes GGA1, GGA7, and GGA20, respectively. After genotyping the external reference samples and combining them to create the external, combined, and diverse panels, each panel was filtered before being phased. As a consequence, the filtering step decreased the number of SNPs by $13.83 \pm 1.36\%$ for the external and combined, and by $23.80 \pm 0.99\%$ for the diverse, which yielded panels with different numbers of SNPs (**Figure 2**A). Regarding the percentage of imputed variants in the 12 validation samples, more than 96% of the total SNPs in each panel successfully passed the multiple filters of the pipeline, even for samples with less than 1× coverage (Figure 2B). The proportion of imputed SNPs increased and gained uniformity across samples when the panel was larger but had fewer SNPs. The mean number of imputed SNPs across samples differed between all the panels: I versus E (T = 14.58, $p<0.001$), E versus C (T = 13.56, $p<0.001$), and C versus D (T = 11.63, $p<0.001$). The $F$-test was significant only between the diverse and the rest of the panels: I versus D (F = 30.54, $p<0.001$), E versus D (F = 24.24, $p<0.001$), and C versus D (F = 11.31, $p<0.001$). Results indicate that the variance across samples for the diverse panel greatly decreased compared to the rest (Figure 2B).

For each imputation scenario, genotype concordance, and precision of het. sites were assessed in the validation samples by comparing imputed and true genotypes per individual. After performing LOOCV with the four reference panels, average values of genotype concordance exceeded 0.90 for every panel (**Figure 3**A) and precision of het. sites ranged from 0.78 to 0.91 (Figure 3B). According to Kruskal Wallis tests, the values of concordance ($p_I>0.85$, $p_E>0.85$, $p_C>0.95$, and $p_D>0.95$), and precision of het. sites ($p_I>0.95$, $p_E>0.85$, $p_C>0.85$, and $p_D>0.85$) did not differ across chromosomes. However, mean values differed between panels for each chromosome (Figure 3). Concordance values significantly differed when comparing the internal, external, and combined panels (Figure 3A). However, no significant differences were detected between the combined and the diverse panels, indicating that imputation accuracy in terms of overall concordance does not increase by adding more distant individuals. For precision of het. sites, differences were detected for all

**2100065 (5 of 14)**

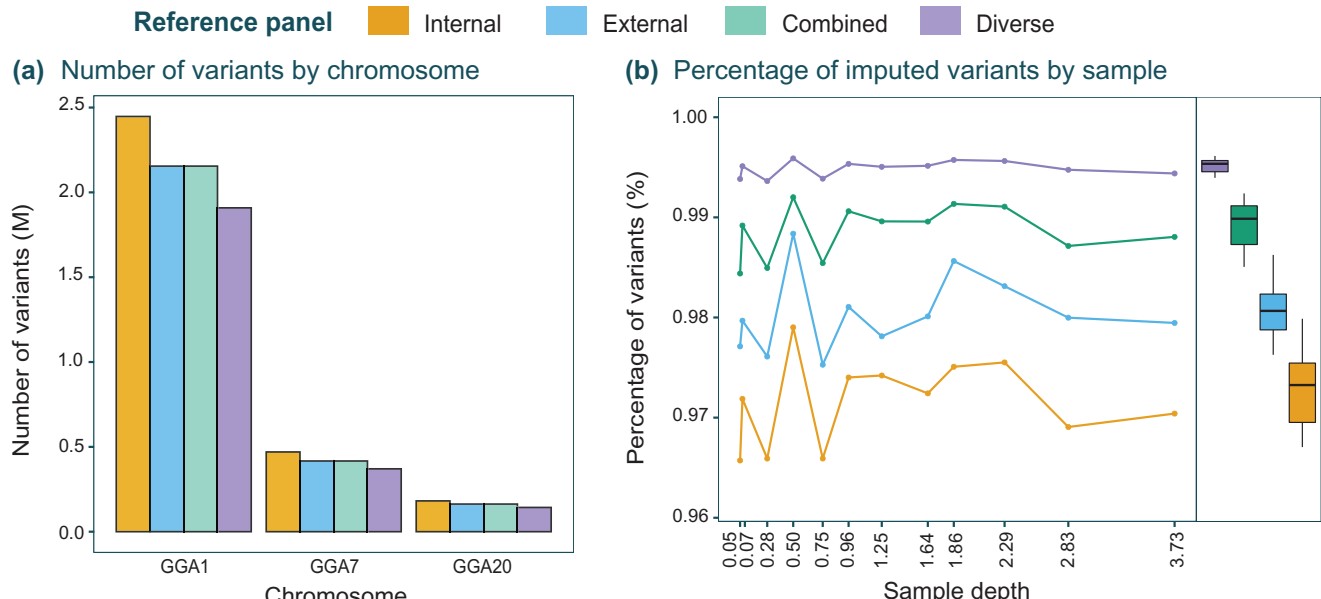

**Figure 2.** Imputation statistics. A) Number of SNPs in each reference panel for chromosomes GGA1, GGA7, GGA20. B) Depth of coverage and proportion of successfully imputed variants of the 12 validation samples for the three chromosomes tested. Capitalized letters refer to panel names: I = internal, E = external, C = combined, and D = diverse.

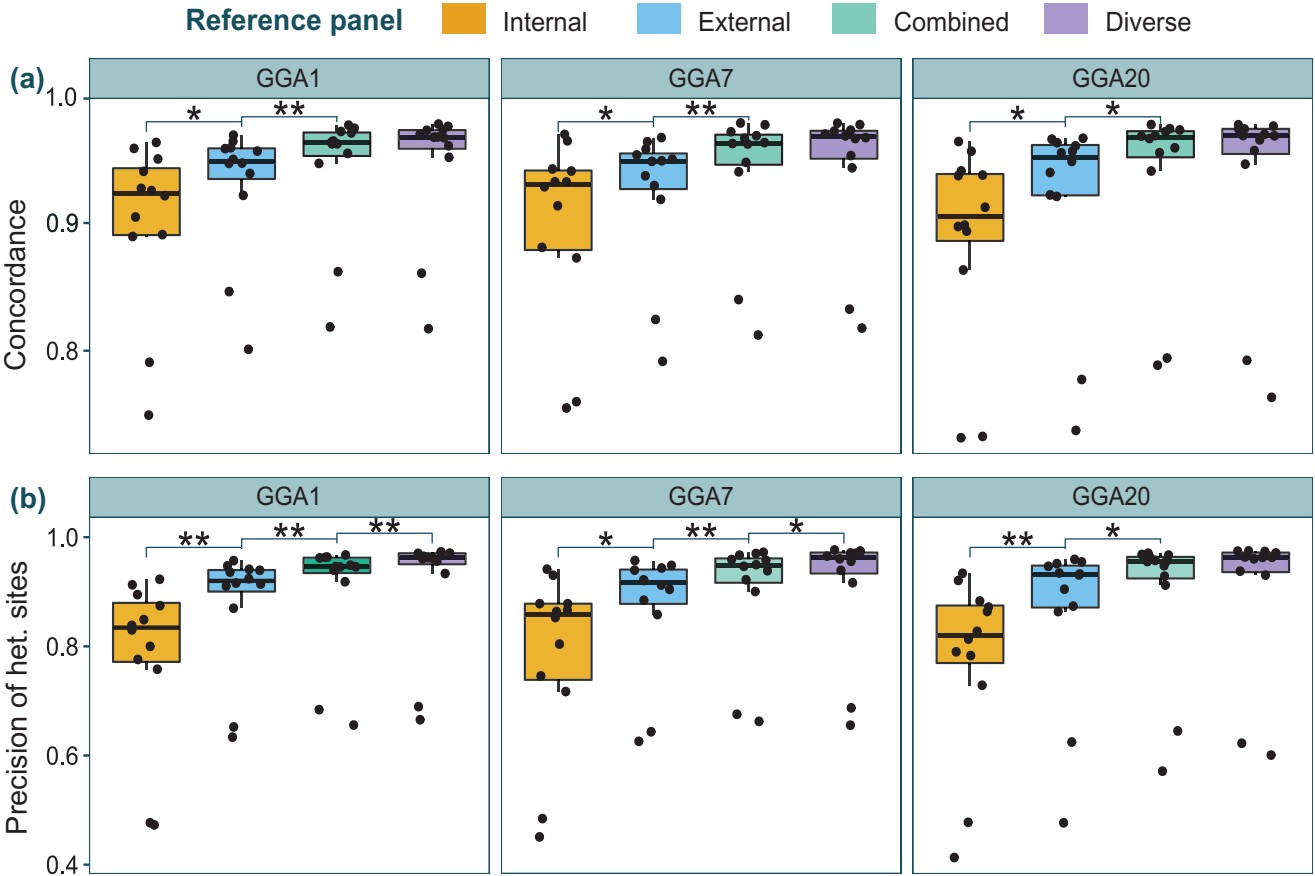

**Figure 3.** LOOCV test results and comparison of imputation reference panels. A) Genotype concordance, and B) precision of heterozygous (het.) sites between imputed (low-depth 12 validation samples) and true (internal reference samples) genotypes on chromosomes GGA1, GGA7, and GAA20. Paired *T*-tests were performed to identify significant differences in means: the following symbols ("**," "*") indicate different *p*-value cut-points (<0.001, 0.05).

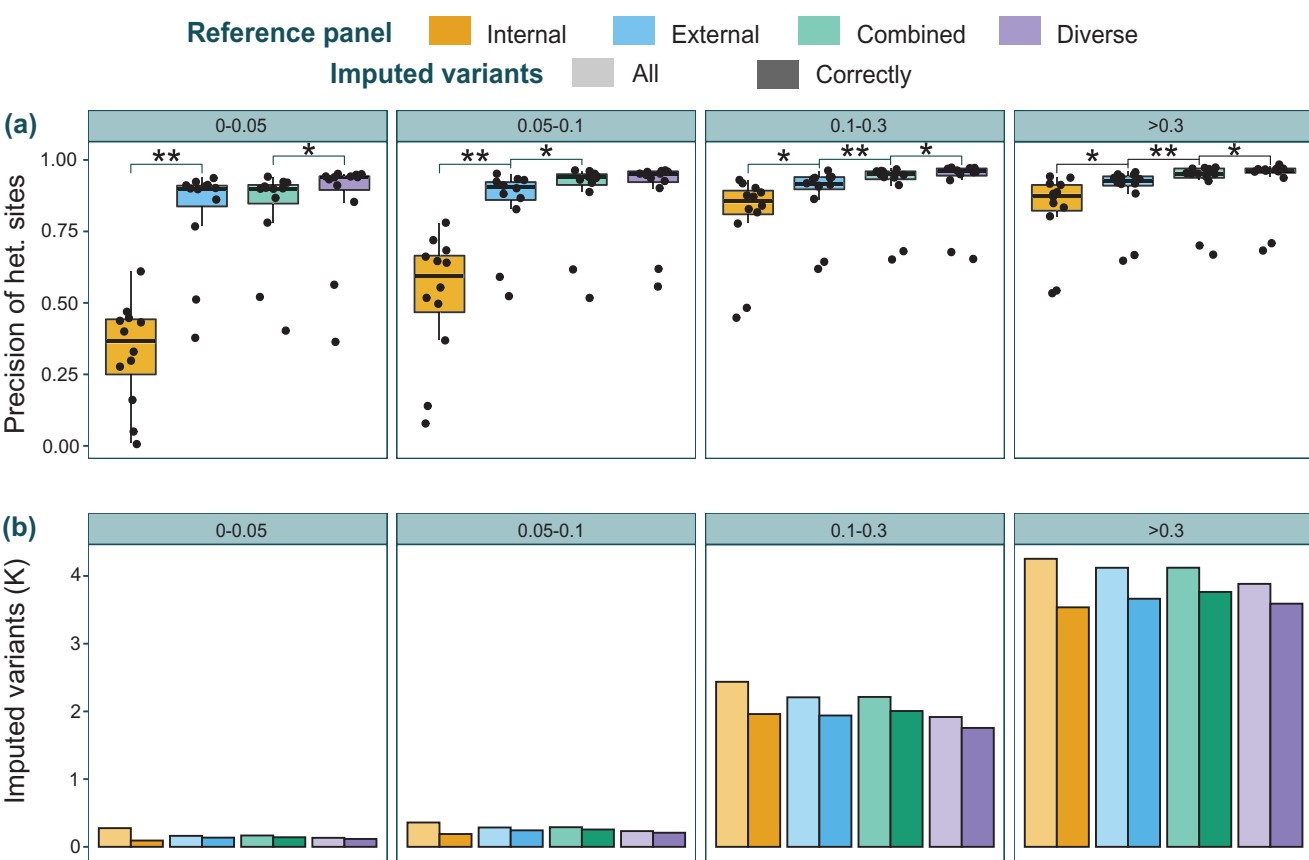

**Figure 4.** Minor allele frequency variants of LOOCV test. A) Precision of heterozygous sites, and B) number of imputed low-frequency variants for chromosome one (GGA1) divided into four different bins of minor allele frequency ranges: 0–0.05, 0.05–0.1, 0.1–0.3, and >0.3. The lower bars represent correctly imputed variants, while the bars with greater transparency represent the number of all imputed variants within the respective MAF bin. Variants that coincided between imputed (low-depth 12 validation samples) and true (internal reference samples) genotypes were considered correctly imputed variants. Paired $T$-tests were performed to identify significant differences in means across panels: the following symbols ("**," "*") indicate different $p$-value cut-points (<0.001, 0.05).

panels (Figure 3B), including for the combined and the diverse, except for GGA20. This suggests that the het. positions are the most sensitive to the imputation process.

In an attempt to further assess imputation accuracy, we classified variants according to their MAF in four bins (0–0.05, 0.05–0.1, 0.1–0.3, and >0.3) and calculated precision of het. sites, and the number of correctly imputed variants for GGA1 (**Figure 4**). The internal panel, while recovering the largest number of variants, was also the panel with the lowest performance in adequately inferring low-frequency variants, especially for the variants with MAF <0.1 (Figure 4A). Although there was no improvement from the external to the combined panel for the smallest MAF bin, a substantial improvement was seen for the rest of the bins. Some significant differences but not as pronounced were also observed when switching from the combined to the diverse panel. Therefore, the combined panel showed overall the best results with the highest number of correctly imputed variants in all MAF bins (Figure 4B), while maintaining a very similar number of imputed SNPs as the external panel. The diverse panel inferred fewer low-frequency variants, but did so more effectively (Figure 4).

Despite the high overall imputation accuracy, the two samples with depths of 0.05 and 0.07× were outliers that did not achieve a sufficiently high concordance (>0.90) and precision (>0.75) with any of the panels and chromosomes (Figure 3). They were thus excluded from the target population, and we refer from now on to 10 validation samples instead of 12.

### 3.3. Panel Choice Impact on Population Genetic Inference

#### 3.3.1. Number of Variants and Their Allele Frequency Distribution in the Imputed Target Population

The final number of SNPs recovered from all autosomal chromosomes in the target population with different panels decreased as more distant individuals were included (**Figure 5**A). This was due to the missing call rate (MCR) filter. Using the internal panel, we recovered 11.7 M filtered SNPs in the target population. These were 30% more recovered variants than when using the diverse panel (8.9 M). Most of the excess variants from the internal panel are low-frequency variants that cannot be confidently recovered

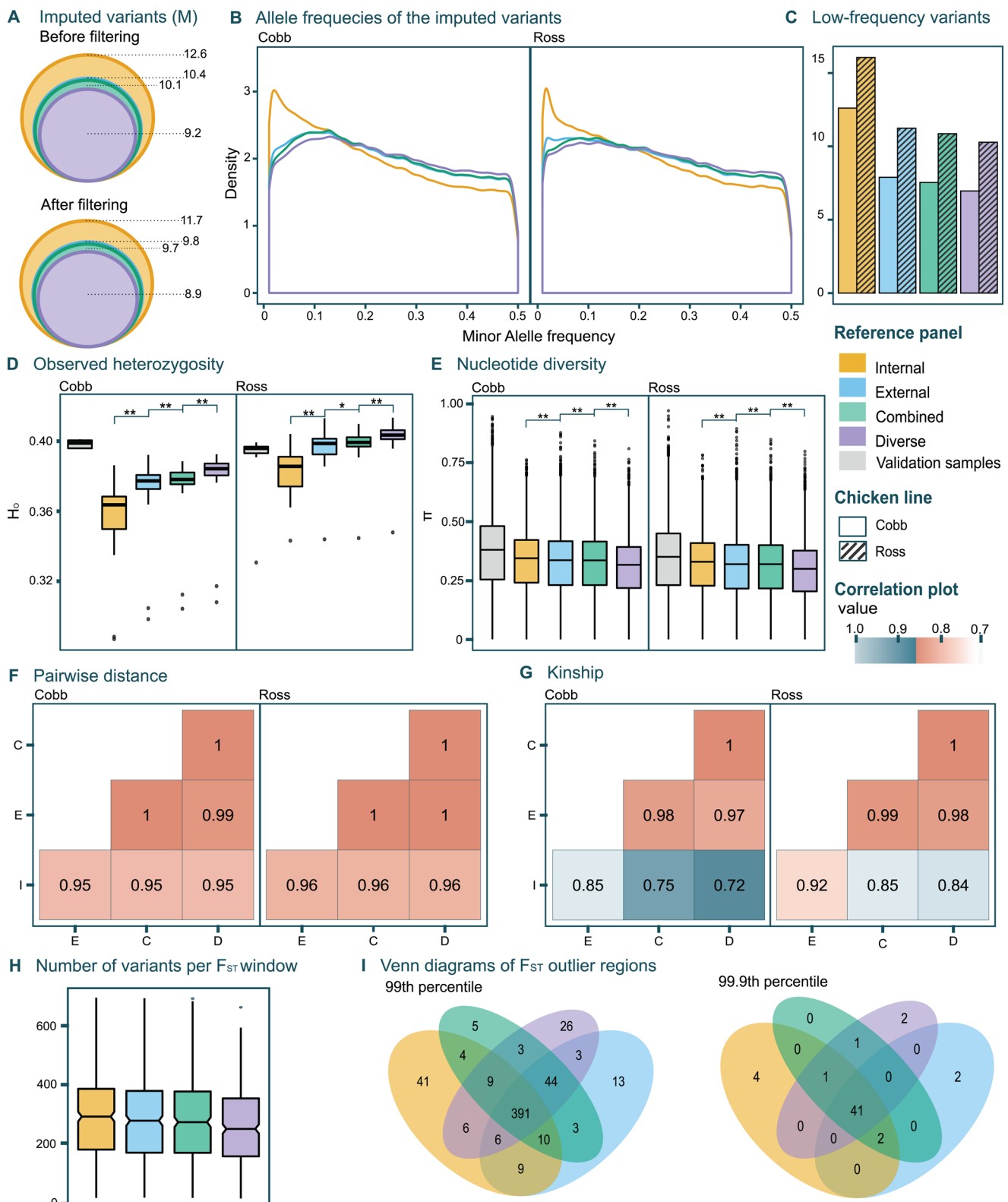

**Figure 5.** Comparison of the choice of reference panels for imputed target population for all autosomal chromosomes. A) Number of variants in the target population when imputed using the different panels. B) Minor allele frequency spectrum of the imputed variants in Cobb and Ross populations. C) Percentage of variants with a MAF lower than 0.05 by broiler line for all the panels. D) Observed heterozygosity for the 10 validation samples (true genotypes) and for the imputed target population by chicken broiler line (Cobb and Ross). Capitalized letters in the legend refer to the following names: I = internal, E = external, C = combined D = diverse, and V = validation samples. E) Nucleotide diversity of the target population by chicken broiler

(Figure 5B), as seen in the lower effective imputation of low-frequency variants with the internal panel (Figure 4A). Both Cobb and Ross populations showed similar allele frequency profiles, with a high proportion of intermediate MAF (Figure 5B) revealing a substantial loss of rare alleles in the respective populations.

### 3.3.2. Population Genetic Inference of the Target Population

Mean Ho values differed across all panels for both Cobb and Ross (Figure 5D). The values estimated by imputation tended to increase with panel size and diversity for both broiler lines. Individual Ho values displayed a higher variance when imputed with the internal panel and tended to equalize across samples with the rest of the panels, following the same trend as with the accuracy statistics (Figures 3 and 5D). This high variance displayed by the internal panel might stem from the fewer correctly imputed variants. For the Cobb population, none of the panels reached the Ho values seen with the 4 Cobb individuals (from the high-depth validation samples) (Figure 5D). For Ross, on the contrary, the external, and combined panels showed very similar values to the validation samples, while the diverse panel overestimated them. The very same trend could be seen when comparing imputed and high-depth validation samples (Figure S3, Supporting Information). There were some outlier samples (two from Cobb and one from Ross) that presented lower Ho than the high-depth validation samples (Figure 5D). These samples apparently underwent an incorrect imputation process, but it was not necessarily related to a low mapping depth.

Nucleotide diversity on the other side, decreased with increasing panel size and diversity (Figure 5E), which was directly related to the lower number of variants retained in the rest of the panels compared to the internal. There were significant differences in means except between the external and combined panels for both genetic lines, most likely because of the similar number of variants both panels share (Figure 5A). When comparing the imputed population with the validation samples, $\pi$ of imputed samples and of the target population were underestimated for all panels (Figure 5E and Figure S3, Supporting Information).

Cobb and Ross populations were very similar, but the imputation tended to accentuate differences between both (Figure 5, Figures S4 and S5, Supporting Information). Within population, pairwise distance and kinship estimates did not vary much according to the panel. For pairwise distance, the diverse panel resulted in larger interindividual distances within genetic lines (Figure S4, Supporting Information). Kinship estimates were lower when computed with the internal panel, since a larger number of SNPs were retained, in particular, low-frequency variants, which are typically unique to one or few individuals, thus decreasing kinship (Figure S5, Supporting Information). Mantel tests did not show any significant differences for pairwise distance and kinship matrices, giving the same result for all panel comparisons (Mantel statistic, $p$-value $< 0.001$). Correlation values for pairwise distance were very similar and close to 1 (Figure 5F), even for the validation samples (true genotypes) when compared with any panel (Figure S3, Supporting Information). For kinship instead, it seemed that the internal panel differed more from the rest (Figure 5G and Figure S3, Supporting Information). In both cases, the 10 validation samples were most correlated with samples imputed with the combined and diverse panels (Figure S3, Supporting Information).

Whole-genome mean $F_{ST}$ values between Cobb and Ross populations were very similar (I = 0.071, E = 0.071, C = 0.072 and D = 0.072) indicating overall low differentiation between the genetic lines. When analyzing the putative selective sweep regions using as threshold the 99th percentile, 68.2% of the windows coincided across the four panels, but more interestingly, 75.9% of the windows were shared across the external, combined, and diverse panels. When we raised the threshold to the 99.9th percentile, 77% of windows were identified by the genome-scans regardless of the choice of panel, indicating that the strongest signals are detected with any panel. Yet, there were some regions that only passed the threshold when imputation was performed with a particular panel (Figure 5I). The combined panel did not show specific sweeps when the percentile was set at 99.9, and it was the panel with the lowest panel-specific regions with the 99th percentile as well, potentially indicating the most robust results, that is, without panel-specific biases. Surprisingly, the diverse panel detected the most panel-specific sweeps after the internal panel (Figure 5I). On the other side, in terms of density of variants in the common windows, the mean number of variants reduced significantly from the internal to the diverse panel (Figure 5H). This suggests that although in a broad sense the same outlier $F_{ST}$ regions tend to be recovered by all panels, a reduced number of imputed variants might decrease the probability of detecting true outliers.

## 4. Discussion

Shotgun metagenomic datasets of host-associated microbial communities often contain host DNA that is usually discarded because the amount of data is too low for accurate host genetic analyses. Here, we introduced an effective and accurate approach to recover high-quality host genomes from gut metagenomic data, which can be used to study host population genetic analyses and ultimately contribute to a better understanding of host-microbiota interactions.

Our analyses yielded drastic differences in mapping statistics between cecum samples used to characterize the target population and ileum samples employed to generate the internal panel. Although both sample types were derived from gut contents, the cecum harbored a very small amount of the host DNA compared to the ileum, because the latter is known to contain fewer bacteria,[45] and a higher permeability and thinner mucus layer of the ileum probably entails higher release of epithelial cells to

line. Paired T-tests were performed to identify significant differences in means: the following symbols ("**," "*") indicate different $p$-value cut-points (<0.001, 0.05). F) Kinship and G) pairwise distance correlation matrices for the target population. Capitalized letters in the $x$– and $y$–axes refer to panel names: I = internal, E = external, C = combined and D = diverse. H) Boxplot showing number of variants in the common windows of the 99th percentile from the $F_{ST}$ genome scan. I) Venn diagram depicting overlap of significantly differentiated windows as estimated by $F_{ST}$ genome scans between Cobb and Ross populations using the different panels for imputation. Significance thresholds were set at the 99th and 99.9th percentiles.

the lumen.[46] Moreover, the low, yet variable, proportion of host DNA retrieved from cecum samples renders sequencing depth adjustment highly unpredictable, as previously reported.[6] Nevertheless, we showed that if a proper reference panel is designed, the low and variable fractions of host DNA recovered from such suboptimal samples can be used for accurately inferring host genetic features.

### 4.1. Adjustment of the Imputation Strategy for Metagenomic Samples

The two-step imputation strategy performed efficiently despite the structural (e.g., study design, animal taxa, reference panel size) differences between our study system and the ones the strategy was originally designed for.[21,22] First, the data pre-processing steps were adapted to the characteristics of the metagenomic samples. As metagenomic data contain multi-species sequences, mapping seed length was increased in order to increase mapping specificity,[29] although there are also examples where they have set standard parameters for the alignment.[2,47] With standard (19) and increased (25) seed lengths, mapping gaps across the reference genome were unevenly distributed. This is evidenced by the large difference between depth (K19 = 2.78x, K25 = 1.8x) and breadth (K19 = 57%, K25 = 50%) of coverage (Figure S1 & Table S1, Supporting Information), likely hampering accurate computation across the genome. However, in our study, accuracy only dropped significantly in samples below 0.1×, while more than 0.90 of the variants were recovered. Similarly, Hui et al. reported that the proportion of correctly imputed heterozygous sites started decreasing at 0.5× of depth of coverage, reaching 50% of correctly imputed sites at 0.1×.[22] Homburger et al. also reported that imputation performance decreased for samples below 0.5× of coverage, being imputation r2 = 0.90 for samples with 0.5× of coverage.[21] Regarding the generation of genotype likelihoods, results should be minimally affected by the choice of the variant caller.[22] We also verified that the DNA obtained from cecum samples was not partially degraded (i.e., short read length, deamination in last bases of the read). Misincorporations at the 5′ (C to T) and 3′ (G to A) ends due to deamination were less than 0.6% (mean = 0.4 ± 0.03%), and read length was consistent across samples (150 bp) (Tables S2 and S3, Supporting Information), thus, we did not apply a deamination filter.

Second, we used custom reference panels with less than one hundred individuals, while the two-step strategy was originally tested with a large human reference panel (i.e., 1000G).[22] Nevertheless, the accuracy of imputed low-frequency variants for all panels was comparable to Hui et al., most likely because the individuals in our target population were closely related as evidenced by the high kinship estimates, and the stringent genotype filtering (MCR = 0) we used for generating the custom panels in order to reduce the error rate.

Finally, unlike humans, avian genomes present macro- and micro-chromosomes, and the latter show higher interchromosomal interactions and recombination rates.[48] However, it seems that the possible crossovers did not affect imputation in contrast to previous studies,[49,50] since we did not find any significant differences in imputation accuracies between chromosomes. This suggests that the strategy worked equally well for large, mid-sized, and small chromosomes with potentially different linkage patterns.

### 4.2. Effect of Reference Panel on Accuracy Statistics

Reference panel design depends on data availability as well as computational capacity. It is a common strategy for imputation of inbred populations to resequence a subset of samples with higher resolution in order to optimize imputation performance.[35] Based on previous works, we estimated that 12 individuals out of 100 would be sufficient to represent the genetic diversity of the population. For instance, previous chicken studies deep-sequenced 25 individuals to impute ≈450 chickens genotyped with 600 K SNP arrays (≈5% of sample size).[18,51]

In terms of the panels' SNP density, we decided to genotype variants that did appear in our target population rather than calling specific variants in the rest of the breeds that composed the reference panels. Thereby, we aimed at reducing the noise that the excess variant density could cause in the imputation process. Nevertheless, as the genetic distance between the reference chicken populations and our broilers is very small,[52] we expected them to share many variants, as we evidenced in preliminary analyses (Figure S2, Supporting Information).

The internal panel resulted in a larger variance across samples for overall accuracy statistics. In addition, SNPs with low MAF had the lowest accuracy when imputed with the internal panel, but were also the poorest imputed across panels. However, incorrectly imputed low-frequency variants can be easily removed if a strict MAF filter is applied for downstream analysis. Another possible option is to sequence more individuals of the target population to increase the reference panel size. Hence, despite the internal panel only representing a small subset of the target population, and showing lower imputation values than the rest of the panels, for scientists without access to external reference samples, this approach is equally useful as overall imputation accuracy was higher than 0.90. In this sense, host resequencing of a small subset of the target population might represent a cost-efficient option, especially for researchers working with non-model organisms and inbred populations.

Our results showed that the combined panel performed better in terms of overall accuracy, and - specifically for MAF variants - than the internal and external panels alone. Despite the fact that the external and combined panels had the same number of SNPs, including a subset of individuals from the target population was beneficial. Many studies already mentioned an improvement for the combined option.[53,54] Last, the diverse panel showed the highest values of concordance and precision of het. sites, most probably because of the lower number of SNPs recovered, especially low-frequency variants, which generally yielded lower imputation accuracies. In terms of imputation of low-frequency variants, the combined panel outperformed the diverse one, that is, it correctly imputed a larger number of variants and tended to improve the precision of het. sites in some MAF bins. A recent large-scale study performed in a Han Chinese population showed that a Chinese-specific reference panel worked better than frequently used reference panels such as 1000G.[19] Imputation was greatly improved when the reference panel contained a fraction

**Table 1.** Metagenomic datasets and available reference panels. Percentage of host DNA was calculated for the studies that did not mention host mapping percentage. Alignment with standard parameters was performed with 10 samples (marked with "*").

| System | Pop. characteristics | N | Sample types | Host DNA | Metagenomic dataset | Ref. panel | Downstream analysis |
|---|---|---|---|---|---|---|---|
| Buffalo | River, swamp, and hybrid buffaloes | 695 | Gut, intestine, and rectum | <20% | [65] | [66] | Selection signatures |
| Cattle | Three crossbreeds and one pureline | 282 | Gut | *3% | [67] | [11] | Selection signatures |
| Pig | Various breeds | 287 | Fecal | 2% | [68] | [69] | Selection signatures |
| Pig | Various breeds | 470 | Fecal | *2% | [70] | [69] | Selection signatures |
| Chicken | Lohmann Brown and Silkie hens | 90 | Fecal | 8% | [71] | Custom[57] | Selection signatures |
| Chicken | Red Junglefowl | 51 | Fecal | 49% | [72] | Custom[57] | Implication on domestication |
| Rat | Sprague Dawley | 49 | Fecal | 11% | [73] | Custom[74] | Host-microbiota association |
| Rat | SpragueDawley | 84 | Cecal | *51% | [75] | Custom[74] | Host-microbiota interactions |
| Mouse | Various breeds | 184 | Fecal | 9% | [76] | Custom[77] | Differences between populations |
| Mouse | C57BL/6J | 88 | Fecal | <5% | [78] | Custom[77] | Differences between populations |
| Zebrafish | Single cohort | 29 | Fecal | *9% | [79] | Custom[80] | Population genetic inference |
| Honey bee | Eastern and Western honey bees | 40 | Gut | <10% | [81] | [82] | Differences between species |

of an extra diverse sample, but they obtained a different pattern when the panel size was fixed.[19] Thus, taking into consideration our and previous results on selection of imputation panels, it can be concluded that increasing panel size and diversity improves imputation, but a balance has to be found in the composition of the panel. The distance between the panel and the target population has to be taken into account.

### 4.3. Effect of Reference Panel on Population Genetic Inference

Besides crude imputation accuracy statistics, we evaluated the impact of the panels on downstream population genetic statistics and their biological interpretation. As imputation accuracies were generally high with our applied pipeline and the stringent filtering approach, we expected population genetic inferences to follow similarly.

Although overall results were in agreement, all the tested parameters showed slight trends according to the used reference panel. Observed heterozygosity, pairwise distance, and kinship values increased while mean $F_{ST}$ and $\pi$ values decreased with panel size and diversity (Figure 5, Figures S4 and S5, Supporting Information). Such biases were related to the composition of the panels and the associated number and distribution of recovered SNPs.

Imputation performance was slightly different for the two broiler lines, as Ross population estimations were closer to the true values than for the Cobb population. Thus, accentuating the distance between both genetic lines. This is most likely due

to a smaller representation of Cobb individuals in the reference panels, that is, 5 Cobb and 7 Ross samples constituted the internal panel. Second, there were some samples that were incorrectly imputed because of their low Ho values (Figure 5D). We do not know if there are individuals with lower Ho in our Cobb and Ross populations. For instance, a Ross individual from the high-depth validation samples had considerably lower Ho than the rest of Ross individuals. Each broiler line came from two different hatcheries, which might be the reason why some individuals might have slightly different genetic features. We may have under-represented one of the origins in the internal reference samples. Thus, it is necessary to be more cautious for the interpretation of individual genomes. Nevertheless, results appeared to be robust and similar across panels at the population level. The genome scans yielded overall very consistent results with major differentiation signals identified by any of the imputed datasets, likely indicative of a true selection signature between both lines. However, downstream analyses, such genome scans, and GWAS must be performed with caution since imputation is sensitive to low-frequency variant quality.

Both broiler lines exhibited high density of intermediate-frequency variants, with similar allele frequency distributions to previously described commercial breed populations.[25,55] A high density of intermediate alleles is indicative of genetic drift due to selection in a closed breeding population.[55] Domestication and breeding history are the two major processes that shape haplotype structure.[25,56] Cobb and Ross, together with other commercial lines, have much smaller effective population sizes than other chicken populations.[57] Broiler breeding methods

are described as a pyramid strategy, in which pure, inbred lines are crossed, then $F_1$ individuals are crossed between each other. In some cases, even a second or a third cross is performed in $F_2$ and $F_3$ generations before raising them for meat.[58,59] Therefore, broilers are highly related populations, but at the same time present high Ho values. Ho of our studied broiler lines was much higher than of local populations,[60] but similar to other commercial lines.[56] Similarly, nucleotide diversity and mean fixation index values were comparable to those previously reported.[25]

### 4.4. Potential Applications

The possibility to retrieve genomic data from metagenomic samples can help reanalyze already published data, and reduce resources spent on population genetic studies. For instance, our approach could be useful to study genome features of endangered populations relying on fecal samples recovered from the environment.[61,62] In an attempt to elucidate the possible applications of the here presented imputation strategy, examples of published metagenomic datasets with potential haplotype reference panels are provided (**Table 1**). As mentioned above, host DNA percentage varies depending on the study system, sample type, and sequencing effort.[63] Some studies collect samples from multiple body sites from each individual,[2,64] and if combined, the amount of host DNA can be greatly increased.[3] Or as in our case, samples with high host DNA can serve as individuals for the reference panel to impute low-coverage samples. Imputation can also be useful for making preliminary explorations of host genome characteristics, while higher quality samples are being processed or sequenced.

The implemented strategy is rather dependent on the availability of a reference genome and high-quality genomic data. Global efforts such as The Vertebrate Genome Project,[83] European Reference Genome Atlas,[84,85] initiatives are contributing to study an increasing number of species, and thereby providing useful reference resources for the scientific community. Likewise, phased haplotype panels are being generated for various species.[69,82] When generating a custom panel in the absence of a publicly available option, the genetic diversity and sample size of the study population have to be taken into account. Larger reference panels are needed with more diverse and heterogeneous populations,[16] while in more isolated populations, reference panels that include population-specific individuals can improve imputation of rare alleles.[19,86,87]

## 5. Conclusion

Our results show that the two-step imputation implemented in this study can be used to successfully reconstruct genotypes and study population genetic properties of hosts from suboptimal metagenomic samples. The comparison among reference panels also demonstrated that this method is versatile and flexible. This approach could be used in many contexts and exploit different data sources to address a variety of research questions. This includes the possibility of mining published metagenomic datasets to recover discarded host DNA sequences. In our particular case, the reconstructed genotypes will be employed in the

H2020 project HoloFood to detect interactions with microbial metagenomic features, and thus implement a hologenomic approach to improve animal production.[88]

## Supporting Information

Supporting Information is available from the Wiley Online Library or from the author.

## Acknowledgements

This research was funded by the European Union's Horizon Research and Innovation Programme under grant agreement No. 817729 (HoloFood, Holistic solution to improve animal food production through deconstructing the biomolecular interactions between feed, gut microorganisms, and animals in relation to performance parameters). The work of S.M. was supported by the Basque Government doctoral fellowship. The authors would like to thank the rest of the HoloFood partners that were involved in the design and execution of the animal trials, specially colleagues Joan Tarradas, Nuria Tous, and Enric Esteve from IRTA.

## Conflict of Interest

The authors declare no conflict of interest.

## Author Contributions

A.E. and A.A. conceived and designed the analyses. S.M. collected the data. S.M. and M.P. performed the analysis. S.M. wrote the original draft. All authors contributed significantly to the review and editing of the original draft.

## Data Availability Statement

Data used in this study is part of the European H2020 reserach project Holofood No. 817729 and will be released upon completion at the end of 2022. Data will be available upon request from the authors until such time.

## Peer Review

The peer review history for this article is available in the Supporting Information for this article.

## Keywords

host, imputation, metagenomic data, population genetics

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

**2100065 (13 of 14)**

Li, X.-M. Lu, E. Lasagna, S. Ceccobelli, H. G. T. N. Gunwardana, T. M. Senasig, S.-H. Feng, H. Zhang, A. K. F. H. Bhuiyan, et al., *BMC Biol.* **2021**, *19*, 118.

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
