## [**Supplementary Information**: Record of Transparent Peer Review · Advanced Genetics]

Recovering high-quality host genomes from gut metagenomic data through genotype imputation

Sofia Marcos *, Melanie Parejo, Andone Estonba, Antton Alberdi*

* Corresponding

Review timeline:	Date submitted:	27-Nov-2021
	Editorial Decision:	07-Feb-2022
	Revision Received:	05-Mar-2022
	Accepted:	01-Apr-2022

Editor: Myles Axton/Kerstin Brachhold

1 st Peer Review	29-Nov-2021 to 07-Feb-2022
----------------------------

Reviewer #1

The manuscript by Marcos et al presents a methodological approach for the analysis of host DNA from gastrointestinal metagenomic datasets. As host DNA content in such datasets is low, the authors present and test a method of imputation to achieve genomic level analysis from the data. The authors establish their technique through analysis of 100 chicken metagenomic gut profiles, and demonstrate the accuracy of their imputation through corroboration of host genotype obtained from high coverage genomic data that is available for 12 of these individuals.

The manuscript is well-written and clear. The caveats of the study are adequately explained. I have few comments to improve the manuscript. The authors could benefit from a more comprehensive justification for the study. They could, for example, cite some publications that make use of large panel metagenomic gut datasets, and use the abundance of this type of data to warrant exploration of their use for additional applications, such as host genomic analyses. In addition, part of the success of the method might be due to the reduced genetic diversity of the broiler population. How might host genomic imputation be affected for other breeds or species where genetic diversity is higher? How applicable do the authors consider this method to be on a broader scale? This is especially relevant given that nucleotide diversity, for what is presumably a rather genetically homogenous population, was underestimated for all panels (lines 409-410). The authors should also note that for their proposed application of using imputation to study endangered species from metagenomic gut profiles is rather dependent on the availability of accurate reference genomic datasets.

The authors should specify somewhere in the abstract that "broiler line" refers to chickens. Not all readers will be familiar with the term.

I also encourage the authors to adjust their last sentence. First, it is an incomplete sentence, but more importantly, whether or not "host contamination" is a problem depends on the research question.

Reviewer #2

The manuscript presented by S. Marcos and colleagues describes a new method for retrieving high quality genotypes from low-coverage host data generated in metagenomic analysis. To achieve this, the authors used an imputation pipeline. This pipeline had as a target a series of samples at low coverage. The genotypes of those targets were estimated using a custom designed panel, which include a subset of target individuals sequenced at higher coverage. The resultant imputed genotypes display a high level of fidelity given a set of population genetic indicator values.

The paper is nicely written and structured, the methodology is well described, and the results seem to be strong. I liked the manuscript and I think it is worth of publication; it would help to reduce the recourses spent in population genetics studies or to reutilise already published data. Nonetheless, I have several doubts I would like the authors to address:

- Do reads characteristics vary depending on the origin of the sample? It is expectable that, since the host DNA is recovered from the intestine, that those sequences are partially degraded (i.e short read length, deamination in last bases of the read)? If so, the variant calling of the target population could have been hampered.

- Have you considered collapsing read pairs before mapping? For what you show in the supplementary figure 1, after increasing the seeding on bwa mem (-k), it seems that you have a substantial number of short reads. Those reads could be too short to be accurately mapped, or worse, could be nonspecific microbial sequences. Could it be possible to remove those sequences given a specific length threshold (i.e, based on the read length distribution after pairs collapse)?

- Regarding the calling, have you considered using angsd (Korneliussen 2014) to do the variant calling instead of bcftools? This software is specifically designed for working with low coverage data and for generating genotype likelihoods.

- My last major comment, and this might be more conceptual. Are chickens (or other livestock for the matter) genetically comparable diversity wise to, let's say, a human or a wild animal? If the overall diversity of the specie analysed is low, you would expect to need less data to successfully impute the variants. Could this explain the discrepancy you expose at the discussion at lines 469 - 471: "Hui et al. documented that the proportion of correctly imputed heterozygous sites started decreasing at 0.5x of depth of coverage, reaching 50% of correctly imputed sites at 0.1x"? I feel that it would be nice to add more emphasis in the abstract and conclusions that results may vary depending on the analysed taxonomic groups or species.

Other minor comments I have:

- On line 466 "depth sequence data, instead of downsampled sequencing reads from high-depth samples. Thus, mapping gaps across the reference genome were unevenly distributed." This could be attributed to read length discrepancy.
- On line 519 "A recent large-scale study performed in a Chinese population showed that a population-specific reference panel worked the best compared to European reference panels such as 1000G." The 1000G is not (exclusively) a European reference panel. Please reformulate or correct the phrase.

1 st Editorial Decision		07-Feb-2022
Editorial Decision: Revise and resubmit after addressing the reviewers' comments		
Editor's understanding of the reviews Reviewer #1 Recommends Minor Revision Reviewer #2 Recommends Major Revision		
Reviewer comments		Editor recommendation
1.1 cite some publications that make use of large panel metagenomic gut datasets, and use the abundance of this type of data to warrant exploration of their use for additional applications, such as host genomic analyses.		ED1 Cite and enumerate the suitable metagenomic sequence datasets (perhaps in a Supplementary Table). In combination with available sequences for imputation (1.2 below) comment on the range of research analyses your method enables..
1.2 part of the success of the method might be due to the reduced genetic diversity of the broiler population. How might host genomic imputation be affected for other breeds or species where genetic diversity is higher? 2.4 Are chickens (or other livestock for the matter) genetically comparable diversity wise to, let's say, a human or a wild animal?... add more emphasis in the abstract and conclusions that results may vary depending on the analysed taxonomic groups or species.		ED2 How dependent is your method on the appropriate host sequence panel? Using the literature and datasets gleaned above, together with availability of diverse high quality sequences for the host taxa studied, comment on the proportion of the available metagenomic datasets that would be addressed by your method.
2.1 Do reads characteristics vary depending on the origin of the sample? It is expectable that, since the host DNA is recovered from the intestine, that those sequences are partially degraded (i.e short read length, deamination in last bases of the read)? If so, the variant calling of the target population could have been hampered.		ED3 For species where you have blood and fecal DNA, how sensitive is variant calling on degradation and deamination of the fecal sample?
2.2 Have you considered collapsing read pairs before mapping? For what you show in the supplementary figure 1, after increasing the seeding on bwa mem (-k), it seems that you have a substantial number of short reads. Those reads could be too short to be accurately mapped, or worse, could be nonspecific microbial sequences. Could it be possible to remove those sequences		ED4 Try this method and report effect on mapping. How else might excess short reads be handled?

given a specific length threshold (i.e, based on the read length distribution after pairs collapse)?	
2.3 Regarding the calling, have you considered using angsd (Korneliusson 2014) to do the variant calling instead of bcftools? This software is specifically designed for working with low coverage data and for generating genotype likelihoods.	ED5 Use alternative calling software and discuss the comparative performance of on your pipeline.

Authors' Response to 1 st Review	05-Mar-2022
---	-------------

Reviewer 1. Comment 1: They could, for example, cite some publications that make use of large panel metagenomic gut datasets, and use the abundance of this type of data to warrant exploration of their use for additional applications, such as host genomic analyses.

EDITOR RECOMMENDATION 1: Cite and enumerate the suitable metagenomic sequence datasets (perhaps in a Supplementary Table). In combination with available sequences for imputation (1.2 below) comment on the range of research analyses your method enables.

RESPONSE 1: Following these suggestions, we have added a new section to the discussion entitled "4.4. Potential applications" accompanied by a table with metagenomic dataset examples [L638]. We give details about the respective available samples for imputation, the proportion of host DNA that samples contain, and suggest possible downstream host genetic analyses.

Reviewer 1. Comment 2: In addition, part of the success of the method might be due to the reduced genetic diversity of the broiler population. How might host genomic imputation be affected for other breeds or species where genetic diversity is higher? How applicable do the authors consider this method to be on a broader scale? This is especially relevant given that nucleotide diversity, for what is presumably a rather genetically homogenous population, was underestimated for all panels (lines 409-410). The authors should also note that their proposed application of using imputation to study endangered species from metagenomic gut profiles is rather dependent on the availability of accurate reference genomic datasets.

Reviewer 2. Comment 4: My last major comment, and this might be more conceptual. Are chickens (or other livestock for the matter) genetically comparable diversity wise to, let's say, a human or a wild animal? If the overall diversity of the species analysed is low, you would expect to need less data to successfully impute the variants. Could this explain the discrepancy you expose at the discussion at lines 469 - 471: "Hui et al. documented that the proportion of correctly imputed heterozygous sites started decreasing at 0.5x of depth of coverage, reaching 50% of correctly imputed sites at 0.1x"? I feel that it would be nice to add more emphasis in the abstract and conclusions that results may vary depending on the analysed taxonomic groups or species.

Editor recommendation 2: How dependent is your method on the appropriate host sequence panel? Using the literature and datasets gleaned above, together with availability of diverse high quality sequences for the host taxa studied, comment on the proportion of the available metagenomic datasets that would be addressed by your method.

Response 2: The reviewers and editor are right that our approach might be particularly successful, most likely due to the close relationship and genetic homogeneity of the individuals in our target population (as already mentioned in the discussion [L527]). This is also evidenced by the fact that even the internal panel, despite having the lowest accuracy values, performed well for population genetic inferences. Nucleotide diversity, as defined by the average pairwise sequence differences per nucleotide site, is directly proportional to the number of variants recovered. It was thus underestimated as more filters were applied in the imputed datasets reducing the number of variants. Nevertheless, diversity as a relative population genetic parameter was higher in all comparisons in Cobb than in Ross, thus keeping its rank. However, the availability of larger reference panels becomes crucial with more diverse and heterogeneous populations. Following your recommendations, we discuss these points in the new section (“4.4. Potential applications”) [L638] highlighting the need of a reference panel and high-quality genomes for our procedure, and how global efforts are being made to generate them. Regarding the table about examples of metagenomic datasets that we added to the discussion [L654], we added haplotype reference panels that are available for each example. Lastly, we also rewrote the abstract.

Reviewer 2. Comment 1: Do reads characteristics vary depending on the origin of the sample? It is expected that, since the host DNA is recovered from the intestine, that those sequences are partially degraded (i.e short read length, deamination in last bases of the read)? If so, the variant calling of the target population could have been hampered.

Editor recommendation 3: For species where you have blood and fecal DNA, how sensitive is variant calling on degradation and deamination of the fecal sample?

Response 2: In order to address this concern, we have now run some tests in our cecum samples to see if there were differences in deamination using DamageProfiler. We see no evidence of DNA deamination nor short read length in the 10 samples tested. Thus, most chicken DNA we recover is most likely not degraded DNA, but consists of intracellular chicken DNA (mentioned in the discussion [L517]). We modified the methods and discussion sections accordingly, and added the results obtained in the comparison to the supplementary information ([L501-521]).

Reviewer 2. Comment 2 Have you considered collapsing read pairs before mapping? For what you show in the supplementary figure 1, after increasing the seeding on bwa mem (-k), it seems that you have a substantial number of short reads. Those reads could be too short to be accurately mapped, or worse, could be nonspecific microbial sequences. Could it be possible to remove those sequences given a specific length threshold (i.e, based on the read length distribution after pairs collapse)?

Editor recommendation 4: Try this method and report the effect on mapping. How else might excess short reads be handled?

Response 3: Thank you for this comment. It was something we were concerned about from the beginning, as there are no standardized alignment methods for multi-species samples. Collapsing reads is an option for very short DNA fragments where paired-end reads overlap in the middle. This was, however, not the case with our sequence data. In fact, during library preparation we included a size verification step using a Fragment Analyzer, and the mean insert size of the paired end reads was 200bp. We based our decision to increase the seed length on the work done by Robinson et al. 2017. According to them, we would be gaining specificity by increasing seed length, but we also found works where standard parameters have been employed for the mapping step (Regan et al. 2018; Blekhman et al. 2015). In our particular case, abnormal alignment statistics and repetitive regions covered by many reads were detected using Geneious and Integrative Genome Viewer (Figure S1). In preliminary analyses (not presented), we tested 19, 23 and 25 seed lengths. The third option gave the best mapping statistics. We modified the third paragraph of the discussion accordingly [L501-521].

Reviewer 2. Comment 3: Regarding the calling, have you considered using angsd (Korneliussen 2014) to do the variant calling instead of bcftools? This software is specifically designed for working with low coverage data and for generating genotype likelihoods.

Editor recommendation 5: Use alternative calling software and discuss the comparative performance of your pipeline.

Response 4: While acknowledging its relevance, as the comparison of different variant calling, imputing and filtering tools have already been covered in Hui's et al (2020) work (who reported a minimal effect of the variant calling strategy), we feel that such an analysis would only create redundant knowledge, and would deviate the main focus of our manuscript, which is on the exploration of reference panels and subsequent analyses to complement and give continuity to the previous work. We have now briefly mentioned the existence of other programs for generating genotype likelihoods in the discussion [L516]:

Minor changes:

Reviewer 1. Minor comment 1: The authors should specify somewhere in the abstract that "broiler line" refers to chickens. Not all readers will be familiar with the term.

R5: We agree with this and have incorporated your suggestion throughout the manuscript.

Reviewer 1. Minor comment 2: I also encourage the authors to adjust their last sentence. First, it is an incomplete sentence, but more importantly, whether or not "host contamination" is a problem depends on the research question.

R6: Yes, we agree. We decided to delete the sentence.

Reviewer 2. Minor comment 1: On line 466 "depth sequence data, instead of downsampled sequencing reads from high-depth samples. Thus, mapping gaps across the reference genome were unevenly distributed." This could be attributed to read length discrepancy.

R7: We decided to delete this paragraph and focus on the fact that metagenomic data have reads from many taxa and that the pre-processing steps, which are not standardized and being more complex, may

interfere. Modifications done related to these sentences are properly explained in the response to the Reviewer 2. Comment 2 and Editor Recommendation 4 section.

Reviewer 2. Minor comment 2: On line 519 "A recent large-scale study performed in a Chinese population showed that a population-specific reference panel worked the best compared to European reference panels such as 1000G." The 1000G is not (exclusively) a European reference panel. Please reformulate or correct the phrase.

R8: We agree on this comment. We corrected the sentence [L589-592].

During the manuscript revision, we decided to make the following changes in order to lighten the reading and bring it into line with the journal's criteria. In addition, all spelling, grammatical and format errors that we detected have been corrected. The mentioned changes have been listed in the following lines in order to be as traceable as possible:

Line Correction

66 Deleted "Due to its lower computational requirements, this approach can be more cost-efficient when studying closely related individuals, such as chickens from a given hatchery. "

85 Deleted "Hence, there are no specific recommendations about the bioinformatic procedures for host genome recovery from metagenomic data sets and the choice of the most optimal panel to maximise accuracy of the imputation process. We also ignore how the choice of a custom reference panel could determine downstream analyses, such as measuring population genetics statistics."

109 Deleted "Our study design involved genotype imputation from four reference panels with different origins and genetic features to a target chicken population characterised through low genomic coverage from intestinal metagenomic data."

275 Added "the percentage of het-sites over the number of variant sites".

276 Added "the average pairwise sequence difference per nucleotide site" and referenced (Nei 1982).

283 Added "Manichaikul et al.'s estimator in" and cited (Manichaikul et al. 2010)."

288 Added "Weir and Cockerham's fixation index (FST) estimate for" and cited (Weir and Cockerham 1984)."

295 Added a new section: "2.6 Statistical analysis".

331 Joined Figures 5 and 6.

331 Illustrated minor allele frequencies in 5B.

625 Added "Both lines exhibited high density of intermediate-frequency variants, with similar allele frequency distributions to previously described commercial breed populations. A high density of intermediate alleles indicates that the genetic drift due to selection in a closed breeding population has a notable effect."

2 nd Peer Review	07-Mar-2022 to 31-Mar-2022
----------------------------

Reviewer #1

The authors have adequately addressed my comments and I recommend the article for publication.

Reviewer #2

The authors have properly addressed and solved all my doubts. I have no further questions and I believe the manuscript is ready for publication.

Final Decision	01-Apr-2022
----------------	-------------

Accept the revised version for publication as the authors satisfactorily addressed the comments of the reviewers.